# Practicing Outdoor Physical Activity: Is It Really a Good Choice? Short- and Long-Term Health Effects of Exercising in a Polluted Environment

**Alessia Tescione *** , **Francesco Misiti** and **Simone Digennaro ***

Department of Human Sciences, Society and Health, University of Cassino and Southern Lazio, 03043 Cassino, Italy
* Correspondence: alessia.tescione@unicas.it (A.T.); s.digennaro@unicas.it (S.D.)

**Abstract:** Background: Air pollution is an environmental risk factor for mortality and the fifth largest risk factor for all causes of death. The practice of regular physical activity is strongly encouraged to achieve a healthy lifestyle. During a physical exercise session, the volume of inhaled pollutants increases. The present study aims to report the evidence about the interaction between polluted air, physical activity, and the interactive effects of these two variables on individuals' health in the light of the significant changes occurring in the daily routine of individuals practicing sport and physical activities after the end of the pandemic. Methods: A mapping review was performed on electronic databases to summarize studies reporting the effects of pollutants on specific health outcomes. A further analysis investigated how physical habits and air quality changed following the COVID-19 pandemic. Results: The current literature suggests that air pollution alters both short- and long-term health outcomes. Nonetheless, exercising is a protective factor against the harmful effects of air pollution. Conclusions: It is necessary for those who train outdoors to evaluate the external environmental conditions. The change should be aimed at improving air quality by implementing stricter legislative guidelines on air pollution thresholds.

**Keywords:** health; air pollution; physical activity; COVID-19

## 1. Introduction

Air pollution is the top environmental risk factor for mortality and the fifth largest risk factor for all causes of death [1]. The World Health Organization (WHO) estimates that air pollution is a major contributor to the global burden of disease, with 9 out of 10 people worldwide breathing polluted air, exceeding the WHO guideline values for ambient air quality [2]. On the other hand, the practice of regular physical activity is strongly encouraged to achieve a healthy lifestyle [3,4]. It has been estimated that physical inactivity is the cause of 3.2 million deaths per year [5]. However, particularly during an outdoor physical exercise session, the volume of inhaled pollutants increases [6]. The urbanization together with economic, social and technological development, decreases air quality and increases sedentary behaviour [7]. Air pollution may impact on physical activity habits and, consequently, limits the benefits of an active lifestyle. It is necessary to investigate the health effects of pollutants during a physical activity session to understand whether the practice of regular physical activity acts as a protective factor against the harmful effects of air pollution.

The comprehension of this phenomenon is further required in the light of the significant changes that have impacted the daily routine of millions of people, such as the effect of the restrictions that were required to reduce the widespread of the COVID-19 pandemic. The confinement affected both psychological and social dimension and people rescheduled their daily routine using new technological tools such as social media to train from home [8].

Several restrictive measures have been adopted by governments and the closure of sports centers led people to choose to train outdoors, with effects on individuals' wellbeing [9–11].

This mapping review aims to report and comment on: (1) the changes in physical activity and sedentary behaviour following the COVID-19 pandemic; (2) short- and long-term effects of air pollutants on specific health outcomes. These effects have been reported with the purpose of highlighting the gaps present in the literature and proposing strategies that involve the area of research, individuals, and governments to mitigate the harmful consequences on the health of those who exercise outdoors.

## 2. Materials and Methods

The present mapping review [12] focused on the scientific evidence concerning the interaction between polluted air and physical activity as well as the interactive effects of these two variables on individuals' health. In particular, two questions were identified: (1) what are the short- and long-term effects on specific health outcomes of those who train in a polluted environment; (2) what are the major outcomes of studies investigating whether physical activity represents a protective factor against the harmful effects of air pollution. A third line of investigation concerned the changes in individuals' lifestyle following the COVID-19 pandemic. A secondary data analysis [13] was conducted on this topic reporting and commenting on the main evidence of studies and research (e.g., Eurobarometer, Italian Statistic National Office, etc.) aiming at analyzing the public's habits in terms of physical activity and sedentary behaviour. The purpose of this study was to identify key concepts and gaps in the research with the view to reflect on potential strategies and guidelines.

*Search Strategies*

Searches were performed in January 2022. The data collection was conducted on Web of Science, PubMed and Google Scholar databases. No limits on date or type of studies were placed. A search string was made with a combination of all necessary keywords (air pollution, physical activity, air pollutants, health). Inclusion and exclusion criteria have been predefined to investigate all titles, all abstracts and full text of identified records.

Inclusion criteria:

1. Epidemiological/observational studies (case-control, cohort, cross-sectional) that broadly investigated the effects of pollutants on several short- and long-term health outcomes;
2. Studies had to specify the terms "blood pressure", "heart rate variability", "oxidative stress", "forced expiratory volume in one second", "maximal oxygen uptake", "fractional exhaled nitric oxide" as short-term outcomes for investigation;
3. Studies had to specify the terms "respiratory disease", "cardiovascular disease", "metabolic disease" as long-term outcomes for investigation;
4. Studies had to include traffic-related and no traffic-related air pollutants including black carbon, oxides of nitrogen, nitrogen dioxide, particulate matter, sulfur dioxide;
5. Eligible studies included those conducted on both animals models and humans; the decision to consider studies conducted on animals was taken as they revealed evidence that is not available with studies conducted on humans.

Exclusion criteria:

1. Type of studies excluded: reviews and metanalysis;
2. Studies investigating the effects of air pollution on the health of subjects with pre-existing diseases;
3. Studies that did not directly address the association between pollutants, short- and long-term health outcomes and physical activity;
4. Studies conducted in indoor environments. Data and study characteristics from all the final eligible studies were imported into a database including study title, authors, publication year, journal, study design and key-words. Following full-text review, 19 studies were included within the mapping review. A flow diagram is

shown in the Figure 1. The following websites were searched manually: The World Health Organization (https://www.who.int/ accessed on 17 July 2022), the European Environment Agency (https://www.eea.europa.eu/ accessed on 17 July 2022).

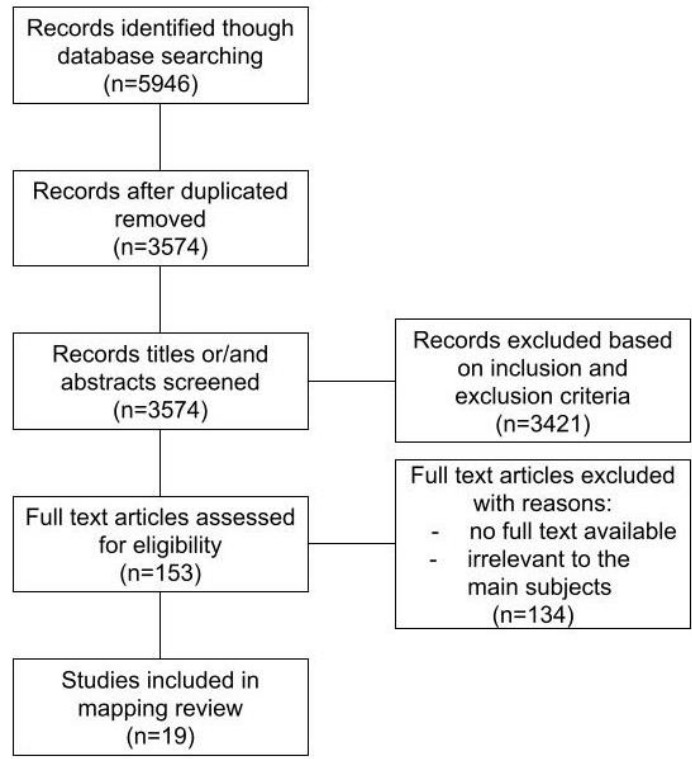

**Figure 1.** Flow diagram describing the selection of studies.

## 3. Results

### 3.1. Impact of COVID-19 on Air Quality, Physical Activity and Sedentary Behaviour

Taking into account COVID-19, this document has been enriched with considerations of the impact of the recent pandemic, as it resulted in a change in air quality, sedentary behaviour and practice of physical activity.

To date, almost three years later, although COVID-19 has not been completely eradicated, it would seem that individuals are gradually returning to their normal daily routines, albeit with changes that have affected both social and personal dimensions [8,14].The role of physical activity is strongly recognized as a means for improving and maintaining good health as well as to prevent the development of various pathological conditions such as cardiovascular disease, metabolic disease, cancer, neurodegenerative diseases, etc. [15–17]. Broadly, sedentary behaviour is extremely high: the WHO, in 2018, reported that more than 1 in 4 adults (1.4 billion people) in the world are physically inactive [2]. This is a truly impressive figure considering that 5 million deaths a year could be avoided if the global population was more active [2]. The implementation of confinement to reduce the spreading of the virus has caused several inconveniences, however the removal of the frenetic rhythms of daily life has allowed everyone more time to devote to themselves. On the other hand, unstructured days may promote obesogenic behaviours [18]. Interestingly, the lockdown has determined significant changes in the way individuals practiced sport and physical activity.

Impact of COVID-19 on Air Quality, Physical Activity and Sedentary Behaviour: A General Overview of the Available Evidence

Between 2020 and 2022, several studies were conducted on the impact of COVID-19 on air quality, physical activity and sedentary behaviour. From the analysis it emerged that the level of physical activity has decreased, with a consequent increase in sedentary

behaviour [19–21]. The reduction of the levels of physical activity affects all ages, without notable differences between males and females. However, it was also found that those who exercised regularly before the pandemic showed a minor reduction in the level of physical activity [10,22–24]. Runancres et al. quantified the change in sedentary time during the COVID-19 pandemic: overall, individuals increased their sedentary time, with children doing so more than adults and without a significant difference by sex [19]. Stockwell et al.'s systematic review summarized the studies that investigated differences in physical activity and sedentary behaviour before and during the COVID-19 lockdown: the 64 studies included in the review reported a decrease in physical activity and an increase in sedentary behaviours during the lockdowns across several populations, including children and patients with a variety of medical conditions [19]. Wunsh et al.'s systematic review and meta-analysis investigated whether and to what extent physical activity changed before and during the COVID-19 pandemic, taking age, sex and measurement method into account. Overall, 37 studies with a total sample size of 119,094 participants from 14 countries worldwide with participants aged between 4 and 93 years were included. Physical activity decreased in all age groups, independently from sex [20].

During the pandemic several European governments encouraged the use of urban and rural green space for practicing physical activity [25]. A recent European survey conducted in 27 Member States of the European Union on a total of 26,578 citizens showed that 18% of respondents stopped being physically active during COVID-19 while 34% claimed to be physically active at the same level as before COVID-19 pandemic [26]. In Italy, the impact of these policies was analyzed with a national survey conducted with a sample of 1503 subjects aged between 18 and 90, representative of the Italian population by gender, age, area of origin and educational qualification. From this research it emerged that 74% of the population practiced physical activity outdoors and only 24% of adults were enrolled in a sports center [27]. Indoor physical activity at sport centers and all sport events have been suspended for a long time, so many people preferred to train at home or practice outdoor physical activity. Between 2020 and 2021, almost all sport centers suffered significant losses in the number of members compared to 2019 [28], with this tendency remaining after the conclusion of the pandemic.

Broadly, outdoor physical activity reduces the risk of contagion but may expose individuals to other health risks caused by exposure to pollutants. COVID-19 had a positive impact on air quality, but this improvement was only temporary; once the global economy recovered from the pandemic, emissions returned to high levels. Several studies have investigated changes in air quality during and after lockdown: some of them compared the lockdown and pre-lockdown period air quality, concluding that there was an improvement during the lockdown period [11,29,30]. Some studies also reported increases for the post-lockdown periods because pollutants' concentration increased to the pre-lockdown levels as soon as the lockdown period ended [31–33].

### 3.2. Practice of Physical Activity: Short- and Long-Term Health Effect of Exposure to Pollutants

Air pollutants are emitted from a range of both anthropic and natural sources. The European Environment Agency categorizes air pollutants in primary, directly emitted to the atmosphere, and secondary sources, formed in the atmosphere from precursor gases through chemical reactions and microphysical processes. Key primary air pollutants include particulate matter (PM), black carbon (BC), sulfurdioxide ($SO_2$), nitrogen oxides (NOX) (including nitrogen monoxide and nitrogen dioxide, $NO_2$), ammonia ($NH_3$), carbon monoxide (CO), methane ($CH_4$), non-methane volatile organic compounds (NMVOCs), including benzene, and certain metals and polycyclic aromatic hydrocarbons, including benzopyrene (BaP). Key secondary air pollutants are PM, ozone ($O_3$), $NO_2$ and several oxidized volatile organic compounds (VOCs). Key precursor gasses for secondary PM are sulfur dioxide ($SO_2$), NOX, $NH_3$ and VOCs [34]. If the practice of physical activity takes place in environments where there is some level of air pollution, exercise increases

the inhalation of pollutants due to the increased frequency and depth of breath [35]. The consequences of the exposure to polluted air may be identified as short- or long-term effects.

### 3.2.1. Short-Term Effects

Short-term effects are acute effects occurring a few hours or days after exposure and are directly measured. The studies included in the present review took into account the following outcomes: blood pressure (BP), heart rate variability (HRV), forced expiratory volume at one second (FEV1), forced vital capacity (FVC), lung inflammation fraction of exhaled nitric oxide (FeNO) and oxidative stress (OS). Based on Kubesh et al. crossover study the exposure to high traffic-related air pollution environment (TRAP) determine higher diastolic blood pressure (DBP) post-exposure, irrespective of physical activity status (rest or intermittent moderate physical activity) and lower systolic blood pressure (SBP) more after exposure to the low TRAP site compared with the high TRAP site [36]. This is confirmed by Kocot et al., who found that DBP showed an increase after exercise and after 15 min of rest [37]. Despite this finding, Kubesh et al. reported that an increase in systolic blood pressure associated with air pollution was attenuated by physical activity [36]. HRV is a measure of the variation in time each heartbeat and a depressed HRV is used to predict the risk of cardiovascular events [38]. In a longitudinal study by Huang et al. the association between particulate matter size and HRV was examined. They found that particles smaller than 0.3 μm may dominate the acute effects of particulate air pollution on reduced cardiac autonomic function [39]. In this regard, Cole-Hunter et al. reported that exposure to TRAP determined a decrease in HRV but physical activity modified the impact especially at high-TRAP sites, acting as a protective factor [40]. The first system that is affected by the effect of pollutants is the respiratory system, even so more during a physical activity session [41]. A cross-over study conducted by Matt et al. tries to disentangle the acute effect of TRAP upon the major respiratory outcomes measured to verify lung function (FEV1, FEV1/FVC and FEF25–75%). They found that physical activity was associated with an increase of FEV1, FEV1/FVC and FEF25–75% and an increase in exposure to one unit (1 μg/m$^3$) of PMcoarse was associated with a decrease in FEV1 and FVC. On the other hand, an increase of physical activity by one unit (Heart rate max) reduced the negative effects of particulate matter on the peak expiratory flow (PEF) and the negative impact of exposure to TRAP constituents on FEV1/FVC and PEF was attenuated in those participants with higher TRAP pre-exposure level [42]. On the same line, Kocot et al. found in their study an acute post-exercise decrease in FEV1/FVC only in participants who exercised under particularly high exposure [43]. In the same study the authors also measured FeNO, a test to determine airway inflammation. They found a decrease immediately after exercise and an increase 15 min later [43]. Physical activity level has been shown to play an important role in the FeNO response; it would seem that those who perform regular physical activity show an increase in FeNO only at a high level of pollution, while those who are inactive show an increase in FeNO even after exposure to low air pollutant concentrations [37]. Histological and biochemical analyses are more sensitive to the harmful effects of low levels of acute PM$_{2.5}$ exposure than physical performance or lung function [44]. Oxidative stress appears to play a key role in the cardiovascular effects of many air pollutants [45]. In this way, recent studies conducted on murine models revealed interesting evidence in relation to physical activity exposure to polluted air and oxidative stress. In particular, Van Waveren et al. showed that there is an increased risk of cardiovascular disease (CVD) associated with exposure to particulate matter, an increased blood pressure and, in this scenario, physical activity during particulate matter exposure provides a protective mechanism. This study conducted on mice models reveals an increase in OS in all groups (control group, physical activity group, particulate matter-exposed group and PA combined with particulate matter exposure group), but at the same time, the groups exposed to physical activity showed an increase in the anti-oxidant system up-regulation, which acts as a protective factor [46]. Moron et al. conducted a similar study but taking into account particle size. Indeed, the study evaluated the influence of both particulate matter PM$_{2.5}$ and

PM$_{10}$ on the oxidative stress parameters. Particle size would appear to be a determining factor; indeed, in groups exposed to PM$_{10}$ (with and without exercise) the antioxidant capacity is better than in groups exposed to PM$_{2.5}$ (with and without exercise) [47]. A further study took into consideration the concentration level of PM$_{2.5}$. The results suggest that a higher concentration corresponds to an increase in oxidative stress, but the level of physical activity modulates this effect. The resistance of tissues to morphological damage is related to the anti-oxidant system up-regulation caused by preventive exercise at each concentration level of PM$_{2.5}$ [48]. A summary of the main characteristics and outcomes of the presented studies is reported in Table 1.

**Table 1.** Summary of studies on short-term health outcomes while being engaged in physical activity in a polluted environment.

| First Author/Year | Population | Main Pollutants | Physical Activity | Health Outcomes | Major Outcomes |
|---|---|---|---|---|---|
| Matt (2016) [38] | 30 healthy adults | PM$_{10}$, PM$_{2.5}$ | Four 2 h exposure scenarios that included either rest or intermittent exercise in high- and low-traffic environments | PEF, FEV1, FEV1/FVC, FEF25–75% | PA was associated with an increase of FEV1, FEV1/FVC and FEF 25–75% and an increase in exposure to one unit (1 μg/m$^3$) of PM$_{coarse}$ was associated with a decrease in FEV1 and FVC |
| Cole-Hunter (2016) [36] | 28 healthy adults | UFP$_s$, BC, PM$_{2.5}$ | 2 h exposure in high or low-TRAP environment. PA consisting of 15 min intervals alternating rest and cycling on a stationary bicycle | HRV | TRAP has an important impact on HRV and that these changes take place within minutes of exposure |
| Kocot (2020) [33] | 76 healthy students | PM$_{2.5}$, PM$_{10}$, SO$_2$, NOx, NO$_2$, NO | Normal training schedule for each participant during physical education classes: volleyball, basketball and judo | FeNO, BP, SpO$_2$ | No significant differences between exposure and control trials in post-exercise BP, HR, and SpO$_2$. In 17 participants, FeNO increased during the exposure trial, while it remained stable or decreased during the control trial |
| Kubesh (2020) [32] | 28 healthy adults | NOX, PM$_{2.5}$ PM$_{10}$, BC | 15 min intervals alternating rest and cycling on a stationary bicycle in high or low-TRAP. | BP | SBP and DBP increase after exposure to TRAP. Intermittent PA attenuated increases in SBP. PM$_{10}$ and PM$_{2.5}$ potentiate these increases |
| Moron (2020) [43] | 48 rats | PM$_{10}$, PM$_{2.5}$ | Aerobic training onthe treadmill, to a moderated intensity, corresponding to 50% of the maximum speed obtained at the test, for 60 min, five times a week, for 4 weeks | Catalase, GPx | The parameters that presented significant differences were catalase and GPx demonstrating an increase in the antioxidant activities in PM$_{10}$ in relation to PM$_{2.5}$. The rats exposed to physical exercise presented a decrease in the anti-oxidant activities compared to the rats not exposed to exercise. |

**Table 1.** *Cont.*

| First Author/Year | Population | Main Pollutants | Physical Activity | Health Outcomes | Major Outcomes |
|---|---|---|---|---|---|
| Van Waveren (2020) [42] | 16 rats | $PM_{10}$, $PM_{2.5}$ | 30 min of moderate intensity exercise per day, five days per week for 8 weeks | SOD, catalase, GPx, FRAP | The PM-exposed rats were hypertensive, showed increased systemic inflammation and oxidative stress. PA was able to decrease systemic inflammation in PM exposed animals, including a reduction in IL-6 serum levels, however, this did not translate to an improvement in BP or vascular reactivity |
| Kocot (2021) [39] | 30 healthy young males | $PM_{10}$, $PM_{2.5}$, $SO_2$ | Two separate 15-min submaximal exercise trials on a cycle ergometer | BP, FeNO | In young and healthy males, exposure to ambient air pollution during short-term submaximal exercise is associated with a decrease in airflow (FEV1/FVC) and the decrease is more apparent when the exercise takes place under particularly high exposure conditions |

BC, black carbon; BP, blood pressure; DBP, diastolic blood pressure; FeNO, fractional exhaled nitric oxide; FEF 25–75%, forced mid expiratory flow; FEV1, forced expiratory volume in one second; FRAP, ferric reducing antioxidant power; FVC, forced vital capacity; GPx, glutathione peroxidase; HR, heart rate; HRV, heart rate variability; $NO_X$, oxides of nitrogen; $NO_2$, nitrogen dioxide; PA, physical activity; PEF, peak expiratory flow; PM, particulate matter; $SO_2$, sulfur dioxide; SOD, superoxide dismutase; $SpO_2$, oxygen saturation; TRAP, traffic-related air pollutants; $UFP_s$, ultrafine particles.

### 3.2.2. Long-Term Effects

Long-term effects studies report health outcomes years after exposure to polluted air taking physical activity into consideration. In particular, for the specific purposes of the present study, the following outcomes have been considered: respiratory disease, cardiovascular disease, metabolic disease. McConnel et al.'s study was based on a cohort of 3535 children (9–16 years old) followed for five years for incidence of asthma. The authors considered participation in physical activity, the number of sports played, the level of air pollution and specific pollutants ($O_3$, $NO_2$, $PM_{2.5}$, $PM_{10}$). They found an increase in the incidence of asthma in children who played more than three sports in areas with high levels of $O_3$ and not in areas with low levels of $O_3$; they also found no difference for the other pollutants [49]. A similar study conducted by Yu et al. measured maximum oxygen uptake ($VO_{2max}$) an indicator of cardiopulmonary fitness, among 821 children (8–12 years old) who lived in high- or low-pollution areas. They gathered information on quality and quantity of practiced physical activity (frequency, duration and level) and on specific pollutants ($SO_2$, $NO_2$, $PM_{10}$). Children who trained in highly polluted areas showed significantly lower VO2max compared to children who trained in low-pollution areas [50]. Fisher et al. measured the incidence of chronic respiratory disease, asthma and chronic obstructive pulmonary disease (COPD) on a cohort of 57,000 adults (50–65 years old). They concluded that cycling, taking part in sports activities and gardening reduce the incidence of asthma and COPD, both in areas with high levels of $NO_2$ and in areas with low levels [51]. Elliot et al. published a cohort study based on 104,990 female participants followed between 1988 and 2008 to measure the associations of 24-month moving average residential, exposure and physical activity updated every four years and the interaction of the two on CVD risk and overall mortality. They found that $PM_{2.5}$ exposure was associated with an increased risk of CVD and overall mortality. A higher level of physical activity was associated with a lower risk of CVD and overall mortality. Lastly, no statistically significant interaction between $PM_{2.5}$ exposure and physical activity in association with CVD risk

and overall mortality was observed [52]. Using a similar approach, Tuet al. followed a cohort of 31,162 individuals (aged 35–74 years) to investigate the interaction among air pollution, physical activity and atherosclerotic cardiovascular disease (ASCVD). The authors evaluated the concentration of air pollutants considering the frequency and the level of physical activity. They concluded that the exposure to high levels of air pollutants were related to increased ASCVD risk, but physical activity attenuated that risk [53]. Long-term exposure to air pollution is linked with metabolic disease. Hou et al. investigated the interaction between air pollution, physical activity, and metabolic syndrome (MetS) following a cohort of 39,089 individuals. The results indicated that long-term exposure to higher ambient air pollutants was related to an increased risk of MetS and the benefit of physical activity decreased with the increase in ambient air pollution [54]. A summary of the main characteristics and outcomes of the presented studies is reported in Table 2.

**Table 2.** Summary of studies on long-term health outcomes while being engaged in physical activity in a polluted environment.

| First Author/Year | Population | Main Pollutants | Physical Activity | Health Outcomes | Major Outcomes |
|---|---|---|---|---|---|
| McConnel (2002) [45] | 3535 children (age range 9–16) | $O_3$, $NO_2$, $PM_{2.5}$, $PM_{10}$ | Number of sports played (0 ≥ 3) | Asthma | Increase in the incidence of asthma in children who played more than three sports in areas with high levels of $O_3$ and not in areas with low levels of $O_3$. No differences for the other pollutants. |
| Yu (2004) [46] | 821 children (age range 8–12) | $SO_2$, $NO_2$, $PM_{10}$ | Frequency, duration and level of physical activity reported by parents | $VO_{2max}$ | Children who trained in highly polluted areas showed a decreased $VO_{2max}$ compared to children who trained in low polluted area. |
| Fisher (2016) [47] | 57,000 adults (age range 50–65) | $NO_2$ | Cycling, sports activities and gardening | Asthma, COPD | Decrease in the incidence of asthma and COPD, both in areas with high levels of $NO_2$ and in areas with low levels. |
| Elliot (2020) [48] | 104,990 female | $PM_{2.5}$ | Physical activity habits measured every 4 years | CVD | $PM_{2.5}$ exposure was associated with an increased risk of CVD and overall mortality. |
| Tu (2020) [49] | 31,162 adults (age range 35–74) | $NO_2$, PM (≤1.0 μm– ≤10 μm) | Physical activity habits measured by International PA Questionnaire | ASCVD | Exposure to high level of air pollution were related to increased ASCVD risk but physical activity attenuated that risk. |
| Hou (2020) [50] | 39,089 adults (age range 18–79) | $PM_{10}$, $PM_{2.5}$, $SO_2$ | Physical activity habits measured by standardized questionnaire | MetS | Long-term exposure to higher ambient air pollutants was related to an increased risk of MetS and the benefit of physical activity decreased by increasing ambient air pollution. |

ASCVD, atherosclerotic cardiovascular disease; CVD, cardiovascular disease; COPD, chronic obstructive pulmonary disease; $MET_S$, metabolic syndrome; $NO_2$, nitrogen dioxide; $O_3$, ozone; PM, particulate matter; $VO_{2max}$, maximal oxygen uptake.

### 3.3. Physical Activity as a Protective Factor against Harmful Effects of Air Pollution

Kim et al. conducted a cohort study on 1,469,972 adults (20–39 years old) to investigate the link between the cardiovascular health benefits of physical activity and the harmful effects of exposure to pollutants during outdoor physical activity. The authors reported a decreased risk of developing cardiovascular diseases in subject with high physical activity

levels ($\geq$1000 min of metabolic equivalent tasks per week) compared to subjects with low levels of physical activity (1–499 MET-min/week) in the condition of exposure to low to moderate levels of $PM_{2.5}$ or $PM_{10}$. One piece of relevant evidence in this study is that subjects with a level of physical activity above 1000 MET min/week who train in the setting of high levels of $PM_{2.5}$ or $PM_{10}$ present an increased risk of cardiovascular disease [55]. A further cohort study conducted by Kim et al. found that moderate or vigorous physical activity ($\geq$5 times/week) was associated with a decreased risk of cardiovascular disease in the setting of both low and high $PM_{2.5}$ [56]. Endes et al. examined physical activity as a modifier of the association between air pollution and arterial stiffness in 2823 adults (50–81 years old), they found that the probability of having an increased risk of arterial stiffness was higher with high $PM_{10}$, $PM_{2.5}$, $NO_2$ in inactive subjects but not in physically active participants [57]. Other studies conducted on the risk of ischemic heart disease and stroke suggesting that physical activity may reduce $PM_{2.5}$ induced risk of cardiovascular events in subjects who are involved in physical activity above 2 times/week [58,59]. Studies conducted on animal models tried to investigate the protective role of physical activity against the harmful effects of air pollution. van Waveren et al. investigated the cardiopulmonary changes in a rat model caused by the exposure of PM during physical activity engagement. The author showed that exercising while exposed to PM provides a protective mechanism against some of the biochemical signalling changes caused by inhaled PM [46]. On the same line, a study examined the effect of an exercise training under exposure to low levels of $PM_{2.5}$ in murine models: in particular, the authors measured heart oxidative stress and the eHSP70 to iHSP70, a biomarker of inflammatory status, and they found that exercise training even at low levels of air pollution may predispose to heart risk caused by an increase in oxidative stress [60].

## 4. Discussion

Air pollution might discourage the practice of outdoor physical activity and reduce the positive benefits of being active. Outdoor exercise and contact with nature has not only physiological but also psychological benefits [61], improving individuals' health, in the broadest sense of the term as defined by WHO [62]. The practice of regular physical activity is strongly encouraged to achieve a healthy lifestyle [4] as low levels of physical activity are associated with several diseases (metabolic disease, cardiovascular disease, cancer) [63]. Hence, physical activity should not be discouraged but exercising with polluted air might be a problem considering that the concentration of air pollutants in many cities is dramatically high. The research community, the general population and local governments must be aware of the effects of air pollution on health with the view to find the right compromise between the promotion of active lifestyles and the excessive exposure to air pollution.

The investigated studies provided interesting evidence, even though they differ in design, time of exposure, type of pollutants, dose of pollutants, and measured outcomes. The studies reported the following overall gaps and weaknesses: (i) doses and particle size are critical factor for estimating potential health risks while exercising, and these information were not always available in the studies; (ii) lifestyle, surrounding house air quality and type of job have an impact on many of the measured outcomes, and this information was not always taken into account; (iii) the current literature does not provide information on air pollution in specific areas dedicated to physical activity; (iv) many of the studies focused on particulate matter ($PM_{2.5}$, $PM_{10}$) without considering the effects of other pollutants; (v) no maximum threshold has been established beyond which the risks exceed the benefits; (vi) additional research is needed to assess the relationship between polluted air and physical activity in the vulnerable population (children, elderly, people with pre-existing diseases); (vii) further studies may investigate the health effects of the interaction between different pollutants and also taking into account environmental conditions such as temperature and humidity.

Regardless of the abovementioned limits, it is safe to conclude that if outdoor sport and physical activities are considered as an important asset for the promotion of individuals'

higher quality of life, specific individual and public measures are required to minimize the risk of exposure to air pollution. Everyone must adopt behaviours that are intended to protect their health, such as: (i) avoiding outdoor exercise near sources of pollution; (ii) avoiding areas with high vehicular traffic; (iii) checking the air quality index of the day; (iv) choosing urban green spaces for working out, as the presence of trees in an urban environment is related to the amelioration of air quality [64]; (v) wearing a surgical face mask as long as it does not compromise sports performance and health outcomes [65].

In addition, there is the need to set specific public policies, with the aim of acting directly on improving air quality and providing information on places, time slots and activities to be preferred. The strategies should also be aimed at improving air quality by implementing stricter legislative guidelines on air pollution thresholds, in line with WHO's recommendation of a maximum threshold value of 5 $\mu g/m^3$ of $PM_{2.5}$ [66], while in Europe the threshold value is set at 25 $\mu g/m^3$ [67]. Local governments should reduce the direct impact on those who train outdoors by implementing air quality monitoring points and make this information available to citizens through, for instance, personal devices. Finally, strategies to mitigate the harmful effects of air pollution should be taken into account in the building of active travel infrastructure: providing accessible urban green areas away from vehicular traffic has a direct impact on health and on promotion of an active lifestyle.

This mapping review offers a broad view of the health effects caused by exposure to polluted air, including multiple health outcomes. However, there are some limitations to be reported: (i) a qualitative analysis of the reported studies was not conducted, as the decision was taken to consider only the effectiveness of the study design; (ii) based on the methodology adopted it was not possible to determine a division by age groups or level of physical activity; (iii) studies conducted in indoor environments were not included, as they differ in terms of the types of indoor pollutants to be analyzed and the type of research designs to be adopted; (iv) studies conducted on subjects with pre-existing diseases were not included as they might be considered a segment of the population that are very susceptible to the deleterious effects of pollutants.

## 5. Conclusions

The current study suggests that physical exercise in a polluted environment contributes to altering health outcomes in the short-and long-term following exposure. On the other hand, it has been shown that physical activity can be a protective factor to counteract the harmful effects of pollutants. Although most available evidence supports the beneficial effects of physical exercise even at the present of higher pollution levels, it is necessary for those who train outdoors to evaluate the external environmental conditions with the view to avoiding excessive negative exposure. Local governments should take into account the changes generated by COVID-19 and implement urban green areas dedicated to leisure/physical activity to have a direct effect on health by reducing the risk of exposure to pollutants and on the other hand making the cities more accessible and sustainable.

**Author Contributions:** Conceptualization, A.T. and S.D.; methodology, A.T., S.D., F.M.; formal analysis, A.T.; investigation, A.T.; writing—original draft preparation, A.T., S.D., F.M.; writing—review and editing, A.T., S.D., F.M.; supervision, S.D., F.M. All authors have read and agreed to the published version of the manuscript.

**Funding:** This research received no external funding.

**Institutional Review Board Statement:** Not applicable.

**Informed Consent Statement:** Not applicable.

**Data Availability Statement:** The authors confirm that the data supporting the findings of this study are available within the article.

**Conflicts of Interest:** The authors declare no conflict of interest.

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
