# Peer review of "Practicing Outdoor Physical Activity: Is It Really a Good Choice? Short- and Long-Term Health Effects of Exercising in a Polluted Environment"

_sustainability, doi:10.3390/su142315790_

Round 1
Reviewer 1 Report
The study suggested that exercise was always a protective factor for our health, regardless of whether the surrounding air is polluted, and the findings further emphasize the importance of exercise. However, the current manuscript still needs some revisions to meet the high-quality standard of the journal.
1. The authors could add a flowchart for selecting and classifying the included articles.
2. The inclusion and exclusion criteria of the included articles (e.g., article types and participants) should be described more clearly.
3. Please provide more information about the process of data synthesis.
4. Please state the rationale for integrating the outcomes of different populations (animals vs. humans).
5. Please state the strengths and limitations of the current study.
6. The title of the manuscript needs to fully reflect the core findings of the research ( air pollution alters both short- and long-term health outcomes nonetheless exercising is a protective factor against the harmful effects of air pollution).
Reviewer 2 Report
Dear authors,
I am pleased to review the paper "Move your Body but Check your Health: The Effects of Air Pollution on Individuals’ Health and how COVID-19 has Changed the Practice of Physical Activity". This is well-structured, clean and transparent paper, it opens a perspective at an intersection of air pollution and physical activity. However, I see a few points on how you can improve your piece of work:
1. Please revise your text and try to simplify it when possible. Due to all the respect to your rich vocabulary and the complexity of the topic, it should be more reader-friendly and easily digestible.
2. (30-31) this statement should be supported by a citation.
3. (39-42) this sentence should be extended and disclosed. I would suggest seeing Glebova, Zare, Desbordes, Geczi (2022) COVID-19 Sport Transformation: New challenges and New opportunities, Physical Culture and Sport Studies and Research, full text available: https://sciendo.com/article/10.2478/pcssr-2022-0011
4. (47-50) this sentence is too heavy for a reader's digestion, could you pleas simplify it by dividing into 2? Thank you.
5. (57-58) it seems a bit confusing and delusive: earlier (48-49) you mention "air quality, the practice of physical activity and sedentary behaviour changed 48 following the Covid-19 pandemic" but right away you rephrase the variables to "polluted air, physical activity" stating that focus is on 2 only variables... It should be cleaner and more logical. Since all these constructs are closely overlapping, it needs an explanation, please.
6. Materials and Methods are not sufficiently explained, please describe the process in detail, step by step. Also, important to know why these methods and sources have been chosen. Please involve more academic literature in this section to show the strengths of your approach.
7. Could you please add the limitations of this study?
Round 2
Reviewer 2 Report
Dear Authors,
Thank you for addressing my suggestions.